# Ceftolozane/Tazobactam for Resistant Drugs *Pseudomonas aeruginosa* Respiratory Infections: A Systematic Literature Review of the Real-World Evidence

**DOI:** 10.3390/life11060474

**Published:** 2021-05-24

**Authors:** Luca Gregorio Giaccari, Maria Caterina Pace, Maria Beatrice Passavanti, Francesca Gargano, Caterina Aurilio, Pasquale Sansone

**Affiliations:** 1Department of Women, Child, General and Specialistic Surgery, University of Campania “L. Vanvitelli”, 80138 Naples, Italy; lucagregorio.giaccari@gmail.com (L.G.G.); mariacaterina.pace@unicampania.it (M.C.P.); mariabeatrice.passavanti@unicampania.it (M.B.P.); Caterina.aurilio@unicampania.it (C.A.); 2Unit of Anaesthesia, Intensive Care and Pain Management, Università Campus Bio-Medico di Roma, 00128 Rome, Italy; f.gargano@unicampus.it

**Keywords:** ceftolozane/tazobactam, hospital-acquired pneumonia (HAP), ventilator-associated pneumonia (VAP), *Pseudomonas aeruginosa*, multidrug-resistant (MDR), extensively drug-resistant (XDR), pandrug-resistant (PDR), carbapenem-resistant (CR)

## Abstract

Background: Ceftolozane/tazobactam (C/T) is a β-lactam/β-lactamase inhibitor combination that mainly targets Gram-negative bacteria. The current international guidelines recommend including C/T treatment in the empirical therapy for hospital-acquired pneumonia (HAP) and ventilator-associated pneumonia (VAP). *Pseudomonas aeruginosa* (PA) is one of the most challenging Gram-negative bacteria. We conducted a systematic review of all cases reported in the literature to summarize the existing evidence. Methods: The main electronic databases were screened to identify case reports of patients with drug-resistant PA respiratory infections treated with C/T. Results: A total of 22 publications were included for a total of 84 infective episodes. The clinical success rate was 72.6% across a wide range of comorbidities. The 45.8% of patients treated with C/T presented colonization by PA. C/T was well tolerated. Only six patients presented adverse events, but none had to stop treatment. The most common therapeutic regimens were 1.5 g every 8 h and 3 g every 8 h. Conclusion: C/T may be a valid therapeutic option to treat multidrug-resistant (MDR), extensively drug-resistant (XDR), pandrug-resistant (PDR), and carbapenem-resistant (CR) PA infections. However, further data are necessary to define the optimal treatment dosage and duration.

## 1. Introduction

Ceftolozane/tazobactam (C/T) is a β-lactam/β-lactamase inhibitor combination that mainly targets Gram-negative bacteria [1]. The U.S. Food and Drug Administration (FDA) in 2014 and then the European Medicines Agency (EMA) in 2015 approved the use of C/T [2,3]. Originally, C/T was indicated for the treatment of complicated intra-abdominal infections (cIAIs) and complicated urinary tract infections (cUTIs) caused by the following Gram-negative and Gram-positive microorganisms: *Enterobacter cloacae, Escherichia coli, Klebsiella oxytoca, Klebsiella pneumoniae, Proteus mirabilis, Pseudomonas aeruginosa, Bacteroides fragilis, Streptococcus anginosus, Streptococcus constellatus,* and *Streptococcus salivarius* [4,5]. The recommended dosage regimen of C/T was 1.5 g administered every 8 h.

C/T was soon investigated as a possible therapeutic agent for nosocomial pneumonia due to its potent activity against *Pseudomonas aeruginosa* (PA) [6], responsible for a large proportion of hospital-acquired pneumonia (HAP) and ventilator-associated pneumonia (VAP) [7,8]. The current international guidelines recommend including C/T treatment in the empirical therapy for HAP and VAP [9,10].

In 2019, the FDA and EMA approved C/T as a therapeutic option for treating HAP and VAP [11,12]. The recommended dosage regimen of C/T was 3 g three times a day (t.i.d.) by intermittent infusion.

PA is one of the most challenging Gram-negative bacteria due to its intrinsic and acquired resistance mechanisms such as its ability to develop biofilms [13]. PA exhibits multidrug-resistant (MDR), extensively drug-resistant (XDR), and pandrug-resistant (PDR) phenotypes, reducing the number of antimicrobial agents available for clinical use [14]. In 2018, the World Health Organization (WHO) selected carbapenem-resistant (CR) PA as a critical-priority bacterium for future development strategies focused on new antibiotics [15].

In this scenario, C/T could play a central role. Ceftolozane is a cephalosporin with potent in vitro activity against PA that is not influenced by the β-lactam resistance mechanisms present in this species [16]. The combination with the tazobactam, a β-lactamase inhibitor, extends ceftolozane activity against many Enterobacteriaceae producing ESBLs [17,18]. According to the Italian survey on PA, C/T was the most active anti-PA agent and, moreover, it was active against approximately half the isolates that are resistant to all other β-lactams or resistant to all other agents except colistin [14].

**Objectives.** We conducted a systematic review of all cases reported in the literature to summarize the existing evidence. To our knowledge, no previously published systematic reviews have evaluated real-world evidence studies of C/T in the treatment of PA resistant to multiple antimicrobial agents. Although a systematic review of case reports cannot support efficacy between drug-resistant PA respiratory infections and C/T therapy, it may identify unrecognized or rare associations and may generate hypotheses for subsequent studies.

## 2. Materials and Methods

### 2.1. Protocol and Registration

A systematic review was performed on the basis of the PRISMA (Preferred Reporting Items for Systematic Reviews and Meta-Analyses) guidelines [19]. The protocol was not published, but is available on request. The review was not registered with the international prospective register of systematic reviews (PROSPERO) due to the long waiting times for the current situation.

### 2.2. Literature Search Strategy

The main electronic databases (Medline, EMBASE, PubMed, Google Scholar, and The Cochrane Library-CENTRAL) were screened to identify case reports of patients with drug-resistant PA respiratory infections treated with C/T. Other relevant studies were identified from the reference lists. We used a combination of terms such as “*ceftolozane-tazobactam*”, “*Pseudomonas aeruginosa*”, “*drug resistance*”, and “*pneumonia*”. The titles and abstracts were screened by two researchers (L.G.G. and M.C.P.) to identify the keywords. The selected papers were read in full by the two independent reviewers, and a third reviewer (P.S.) was consulted in cases of disagreement.

The initial search was performed on 1 April 2021. All publications were included since inception up until the end of March 2021.

All the papers with available full text, reporting original data of patients with drug-resistant PA respiratory infections treated with C/T, of any age, gender, and in any setting, were included. No language restrictions were applied.

### 2.3. Inclusion and Exclusion Criteria

Studies were included if they met all of the following criteria:
−the full study was published;−the study described clinical use of C/T for respiratory infections;−the responsible agent of the infection was PA resistant to multiple antimicrobial agents;−the study reported the clinical outcome of the patient(s) treated with C/T.

Reasons for exclusions were:
−the study did not report clinical outcome;−the study had duplicate data with others (in these cases, only the largest study was retained);−the study presented pooled data that did not allow for extrapolation of useful information.

### 2.4. Data Extraction

Data were independently extracted by one of the three reviewers (L.G.G., M.C.P., P.S.) according to a predefined protocol. The data extraction was then checked by one of the other two reviewers, and the discrepancies were resolved by discussion between all of them.

Variables of interest included:
−demographic characteristics (sex and age);−clinical characteristics (commodities);−type of infection and resistance profile;−therapeutic regimen (empirical and targeted) and dosage;−co-infections;−adverse events (AEs);−clinical and microbiological outcome.

According to an international expert proposal, MDR is defined as “non-susceptibility to at least one agent in three or more antimicrobial categories”, XDR is defined as “non-susceptibility to at least one agent in all but two or fewer antimicrobial categories (i.e., bacterial isolates remain susceptible to only one or two categories)”, and PDR is defined as “non-susceptibility to all agents in all antimicrobial categories (i.e., no agents tested as susceptible for that organism)” [20].

## 3. Results

A total of 22 full texts were eligible for inclusion, namely, 4 cohort studies and 18 case reports/case series investigating 84 infective episodes in total. As shown in Figure 1, the flow diagram reports the results from the literature search and the study selection process.

### Study Characteristics

In Table 1, all the studies are presented in alphabetical order with a brief clinical description for each case.

Five countries were included: USA (12 studies [22,24,28,29,30,32,33,34,35,37,39,40]), Spain (4 studies [23,26,31,41]), Saudi Arabia (3 studies [21,25,42]), the UK (2 studies [36,38]), and France (1 study [27]).

The mean age was 58.89 ± 19.33 years (14–91 years). Sex was reported for 81 patients: males were 58 (71.6%) and females were 23 (28.4%).

The most common comorbidities were respiratory: cystic fibrosis was reported in 13 cases [24,29,35,37,39,40], while COPD was reported in 7 cases [23,26,29,33,38,41]. Ten patients had lungs transplanted [26,27,29,39]. The overall rate of respiratory failure was 17.5%, and nine patients required tracheotomy before developing pneumonia [22,28,33,35,41].

Cardiac diseases were reported in 21 patients, and arterial hypertension was the most common (eight cases [21,23,25,39,41]). Seven patients had diabetes [23,25,26,37,39,42]. A renal impairment was reported in 10 cases: three patients presented with an acute kidney disease [26,32], while seven patients had a chronic kidney disease [21,25,35,37,39].

A history of solid cancer was described in seven patients [25,26,28,33,35,41]. Nine patients had blood cancer [27,30,35,41], and three of them had undergone blood stem-cell transplant [27,29,30].

Immunosupression conditions were reported in seven cases [26,27,28,31,41,42].

Neurological and abdominal comorbidities were also reported. The details are reported in Table 2.

As shown in Figure 2, C/T was used to treat ventilator-associated pneumonia (VAP) in 14 cases [22,23,25,27,32,34], hospital-acquired pneumonia (HAP) in 11 cases [25,26,36,42], tracheobronchitis in 7 cases [29,31,41], and aspiration pneumonia in 1 case [21]. In the remaining 55 patients, the etiology was not clearly defined. Emphysema was associated in three cases [29,35].

MDR was reported in 44 cases [21,26,28,29,30,31,32,33,36,37,39,40,41,42], CR in 25 cases [25,34,35], XDR in 11 cases [27,41], PDR in 4 cases [22,23,38]. Among CR-PA, no cases of carbapenemase-producing Enterobacteriaceae (CPE) were reported.

In 26 patients, a co-infection was detected (see Table 3).

As shown in Figure 3, out of 35 co-infection episodes, 17 microorganisms (49%) were isolated in the lungs, 7 microorganisms (20%) in the blood, 4 microorganisms (11%) in the gastrointestinal tract, and 2 microorganisms (6%) in the urinary tract. Of the five cases reported by *Dinh et al.*, none of the sites of infection were reported.

The most frequent bacterial microorganisms were Staphylococcus aureus (seven cases [24,29,37,38,41]), Pseudomonas aeruginosa (six cases [21,23,27,31,37,38]), Clostridium difficile (three cases [23,28,33]), and Klebsiella pneumoniae (three cases [27,35]). Two cases of fungal infections occurred due to Candida auris [34] and Candida tropicalis [29]. A case of herpes simplex pneumonia was reported by *Álvarez Lerma et al*. Details are shown in Figure 4.

As shown in Table 1, empirical treatment was performed in 40 cases in the absence of microbiological isolation on the basis of clinical or radiological parameters. Targeted treatment consisted of C/T alone or C/T in association with other antibiotics. The most common association was with colistin intravenously injected or inhaled (27 patients [22,25,26,27,29,33,34,35])*,* tobramycin intravenously injected or inhaled (15 patients [21,29,33,35,40,41]), ciprofloxacin intravenously injected (7 patients [24,29,37,40]), amikacin intravenously injected (5 patients [27,41,42]), and linezolid intravenously injected (4 patients [29,38]).

In 31 cases [22,25,26,27,29,31,33,35,36,41,42], CT was administered as 1.5g every 8 h, while in 26 cases as 3 g every 8 h [26,27,28,29,30,32,34,35,37,38,40]. Other C/T regimes were 375 mg every 8 h (seven patients [29,35]), 750 mg every 8 h (six patients [23,35]), 150 mg every 8 h (two patients [29]), 2 g every 8 h (one patient [26]), and 3 g every 12 h (one patient [39]). In *Alessa et al.*, a bolus (1.5 g) was performed before starting C/T 300 mg every 8 h. In three cases [23,24,41], the administration protocol was adapted during the treatment. Three studies (six patients [25,27,41]) did not report the C/T regime.

C/T mean duration therapy was 15.61 ± 13.28 days; it ranged from 3 days [26,29] to 103 days [30].

Only six patients presented adverse events, but this did not lead to the therapy discontinuation. The reported AEs were mild transaminasis [24], confusion/hallucinations, renal failure, anemia [27], leukocytosis [34], eosinophilia/eosinophyluria, and interstitial nephritis (*Munita et al.*).

As shown in Figure 5, the overall success rate of C/T was 72.6% (61/84 cases). Failure was more frequent when considering CR-PA (40%, 10/25 cases), followed by XDR-PA (36.4%, 4/11 cases) and MDR-PA (20.5%, 9/44 cases). Exitus occurred in 13 cases.

After C/T therapy, the eradication of infection occurred in 26 patients, whereas the patients colonized by PA were 22 in number (see Figure 6). In 36 cases, no microbiological outcome data were available.

## 4. Discussion

PA is a challenging Gram-negative bacterium due to its intrinsic and acquired resistance mechanisms and its ability to develop biofilms [13]. To date, the optimal management of the PA infections, especially in cases of multidrug-resistant strains, it is not clarified. First, the superiority of combination therapy over monotherapy as definitive treatment is not established [43]. Current recommendations suggest double anti-PA empirical therapy, especially if resistant isolates are suspected [44,45]. Second, there is no agreement on the antibiotic treatment to choose. C/T is emerging as a good option among β-lactams alone or in combination (especially with colistin or amikacin) for acute invasive PA infections [14,45]. In vitro, C/T is the most active β-lactam against PA, even in the case of resistant strains. C/T is active against 75–89% of the isolates not sensitive to ceftazidime, piperacillin/tazobactam, and meropenem [46].

In our systematic review, the clinical success rate was 72.6% across a wide range of comorbidities. According to the literature, PA causes infections in patients with comorbidities, such as cystic fibrosis, chronic structural lung disease (e.g., bronchiectasis, COPD), impaired immune defenses (e.g., HIV infection, malignancy with neutropenia or recent chemotherapy, organ transplant recipients), diabetes mellitus, renal failure and hemodialysis, and chronic cardiovascular or neurological disease [47]. Notably, clinical success was even higher when considering MDR-PA cases only (83.6%). According to previous data, CR-PA represents a therapeutic problem with a failure rate of 40% in present studies. In this direction, the WHO supports the development of new antibiotics that are active against multidrug-resistant Gram-negative bacteria [15]. The 45.8% of patients treated with C/T presented colonization by PA.

The most common therapeutic regimens were 1.5 g every 8 h and 3 g every 8 h. The latter represents the recommended dosage regimen for treating HAP and VAP [11,12]. Ceftolozane has shown adequate penetration in lung tissue, close to 50%, with 2 g doses in an extended infusion of 3 h [48].

Importantly, C/T was well tolerated. Only six patients presented adverse events, but no one had to stop the treatment. The increased dose of C/T could be associated with an increase in the occurrence of adverse effects, such as elevation of liver enzymes, hypotension, or cutaneous reactions [48].

This study contributes to the accumulation of so-called “real world data”, which, when properly interpreted, can provide suggestions and guidelines [49]. Indeed, due to the recognized gap between the global burden of resistance bacteria and new antibiotics under development, many proposals have been made to optimize the design of clinical trials, but not all recommendations are feasible or immediately applicable [50,51]. In this way, real-world evidence may come as useful.

To summarize, this is the first systematic review of the use of C/T in resistant pulmonary PA infections. Its strength is based on rigorous inclusion criteria, which allow it to collect data exclusively on MDR-PA, XDR-PA, PDR-PA, and CR-PA infections and determine the outcome.

**Limitations.** This study presents some limitations. First, it was based on observational studies, the majority of which were case series or case reports. Due to the greater potential for bias, they are often excluded from systematic reviews of treatments. In this report, case series and observational studies contribute substantially to the available evidence base, and their results supplement the limited evidence available from other studies. Second, a meta-analysis was not performed, owing to the design of most of the studies (case report, case series) and the lack of a comparator.

## 5. Conclusions

C/T may be a valid therapeutic option to treat MDR-PA, XDR-PA, PDR-PA, and CR-PA infections. However, further data are necessary to define its optimal treatment dosage and duration.

## Figures and Tables

**Figure 1 life-11-00474-f001:**
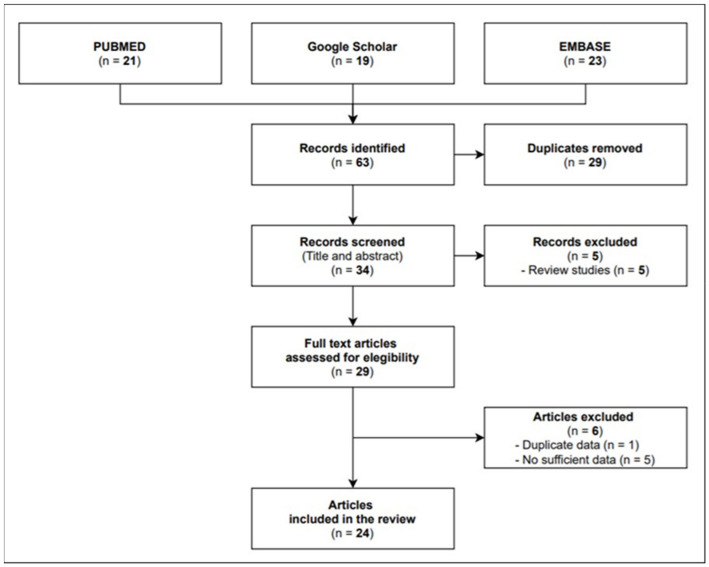
Flow diagram study selection process.

**Figure 2 life-11-00474-f002:**
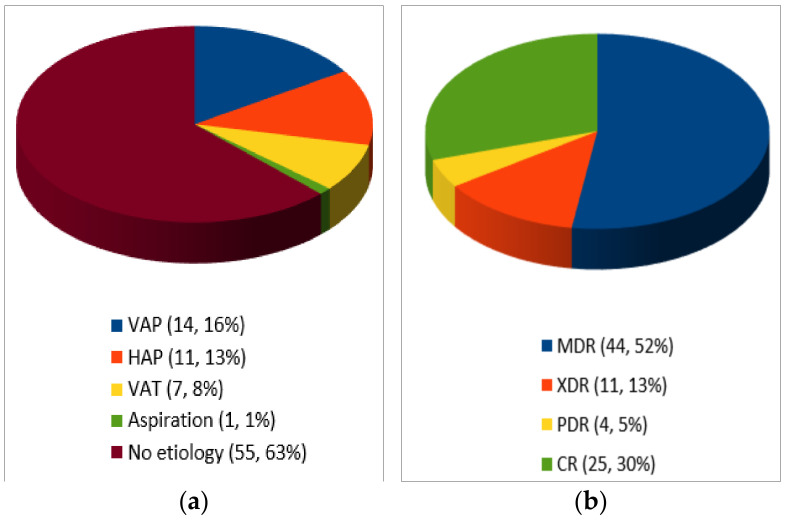
Type of infection (**a**) and resistance profile (*n*, %)(**b**).

**Figure 3 life-11-00474-f003:**
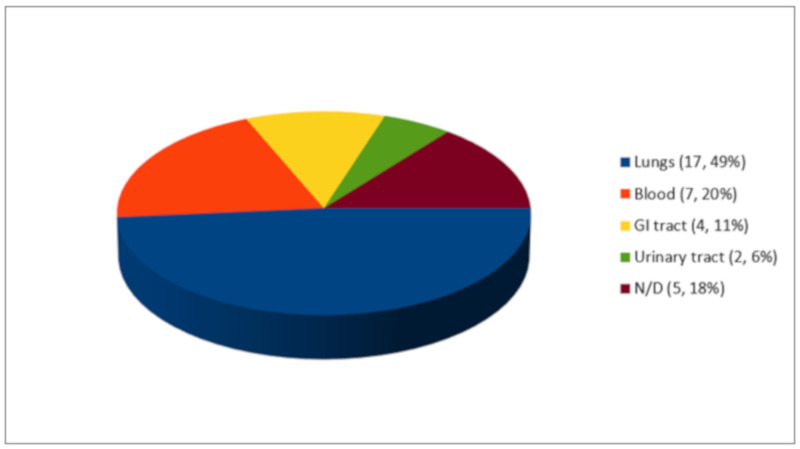
Co-infections (*n*, %).

**Figure 4 life-11-00474-f004:**
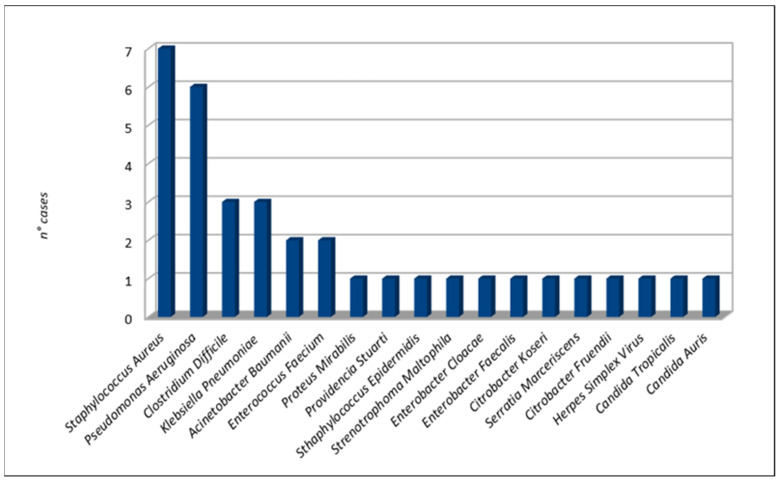
Co-infection microorganisms.

**Figure 5 life-11-00474-f005:**
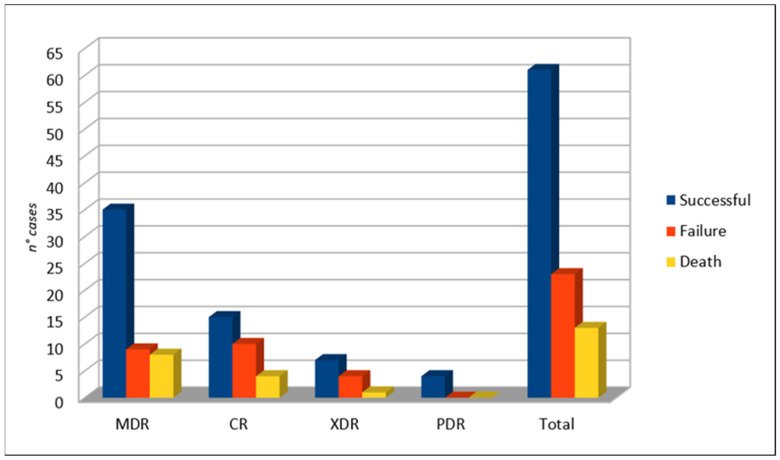
Clinical outcome.

**Figure 6 life-11-00474-f006:**
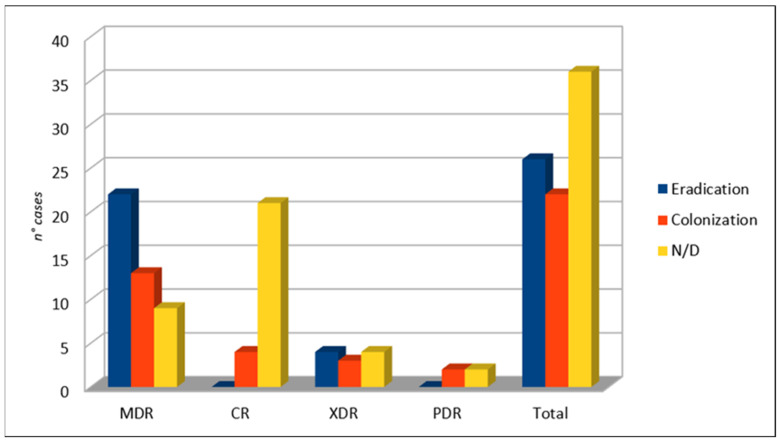
Microbiological outcome.

**Table 1 life-11-00474-t001:** Case reports of PA pulmonary infection.

Author, Year	Study Design	No.	Sex, Years	Type of Infection	Resistance Profile	C/T Dosage (Duration, Days)	Clinical Outcome	Microbiological Status
**Alessa et al., 2018** [21]	Case report	1	M, 79	Aspiration pneumonia	MDR	1.5 g → 300 mg/8 h (13)	S	N/D
**Alqaid et al., 2015** [22]	Case report	1	M, 69	VAP	PDR	1.5 g/8 h (14)	S	N/D
**Álvarez Lerma et al., 2017** [23]	Case report	2	M, 72	VAP	PDR	1.5 g/8 h (4) → 750 mg/8 h (10)	S	C
			M, 48	VAP	PDR	750 mg/8 h (17)	S	C
**Ang et al., 2016** [24]	Case report	1	F, 14	Pneumonia	MDR	1.5 g/8 h → 750 mg/8 h (14)	S	E
**Bosaeed et al., 2020** [25]	Observational study	6	F, 45	Pneumonia	CR	1.5 g/8 h (7)	F	C
			?	Pneumonia	CR	?	S	N/D
			?	Pneumonia	CR	?	S	N/D
			M, 69	VAP	CR	1.5 g/8 h (14)	F	C
			M, 61	VAP	CR	1.5 g/8 h (8)	F	C
			?	VAP	CR	?	S	N/D
**Castón et al., 2016** [26]	Case series	6	F, 75	Pneumonia	MDR	3 g/8 h (10)	S	E
			M, 37	Pneumonia	MDR	3 g/8 h (10)	S	E
			M, 74	Pneumonia	MDR	3 g/8 h (15)	F (death)	E
			M,79	Pneumonia	MDR	2 g/8 h (14)	S	N/D
			M, 61	Pneumonia	MDR	3 g/8 h (3)	F (death)	C
			M, 58	Pneumonia	MDR	1.5 g/8 h (21)	S	C
**Dinh et al., 2017** [27]	Case series	7	M, 61	VAP	XDR	3 g/8 h (15)	S	E
			F, 70	VAP	XDR	3 g/8 h (10)	S	C
			M, 60	VAP	XDR	3 g/8 h (15)	S	C
			F, 73	VAP	XDR	1.5/8 h (18)	F	E
			M, 73	VAP	XDR	1.5/8 h (4)	F	N/D
			M, 49	VAP	XDR	3 g/8 h (11)	S	N/D
			M, 38	Pneumonia	XDR	? (5)	S	N/D
**Gelfand et al., 2015** [28]	Case series	3	M, 69	Pneumonia	MDR	3 g/8 h (14)	S	E
			M, 63	Pneumonia	MDR	3 g/8 h (14)	S	E
			M, 52	Pneumonia	MDR	3 g/8 h (10)	S	E
**Haidar et al., 2018** [29]	Observational study	18	M, 58	Pneumonia, empyema	MDR	0.15 g/8 h (29)	F (death)	C
			F, 23	Pneumonia	MDR	1.5 g/8 h (14)	S	N/D
			F, 84	Pneumonia	MDR	1.5 g/8 h (17)	S	N/D
			M, 70	Pneumonia, empyema	MDR	1.5 g/8 h (14)	S	E
			F, 48	Pneumonia	MDR	1.5 g/8 h (41)	S	N/D
			M, 75	VAT	MDR	1.5 g/8 h (31)	F (death)	C
			F, 55	Pneumonia	MDR	3 g/8 h (42)	S	C
			F, 25	Pneumonia	MDR	750 mg/8 h (52)	S	E
			F, 89	Pneumonia	MDR	750 mg/8 h (14)	F (death)	C
			F, 84	Pneumonia	MDR	375 mg/8 h (3)	F (death)	E
			F, 91	Pneumonia	MDR	750 mg/8 h (10)	S	E
			F, 59	Pneumonia	MDR	1.5 g/8 h (13)	S	E
			F, 41	Pneumonia	MDR	3 g/8 h (14)	S	C
			M, 58	Pneumonia	MDR	150 mg/8 h (15)	S	C
			M, 23	Pneumonia	MDR	1.5 g/8 h (10)	S	E
			M, 39	Pneumonia	MDR	1.5 g/8 h (13)	S	E
			M, 65	Pneumonia	MDR	750 mg/8 h (13)	S	E
			M, 34	VAT	MDR	1.5 g/8 h (4)	S	E
**Hakki et al., 2018** [30]	Case series	3	F, 26	Pneumonia	MDR	3 g/8 h (14)	S	E
			M, 71	Pneumonia	MDR	3 g/8 h (31)	F	C
			M, 54	Pneumonia	MDR	3 g/8 h (103)	S	E
**Hernandez-Tejedor et al., 2016** [31]	Case report	1	M, 58	VAT	MDR	1.5 g/8 h (10)	S	C
**Kuti et al., 2016** [32]	Case report	1	M, 75	VAP	MDR	3 g/8 h (10)	S	E
**Lewis et al., 2018** [33]	Case report	1	F, 53	Pneumonia	MDR	1.5 g/8 h (12)	F (death)	N/D
**Maniara et al., 2021** [34]	Case report	1	M, 25	VAP	CR	3 g/8 h (15)	S	N/D
**Munita et al., 2017** [35]	Observational study	18	M, 31	Pneumonia	CR	1.5 g/8 h (14)	F (death)	N/D
			M, 38	Pneumonia	CR	750 mg/8 h (42)	S	N/D
			M, 16	Pneumonia	CR	3 g/8 h (28)	S	N/D
			M, 32	Pneumonia	CR	1.5 g/8 h (18)	F	N/D
			M, 35	Pneumonia	CR	3 g/8 h (9)	S	N/D
			M, 25	Pneumonia	CR	1.5 g/8 h (8)	S	N/D
			M, 30	Pneumonia	CR	1.5 g/8 h (14)	S	N/D
			M, 26	Pneumonia	CR	1.5 g/8 h (27)	F (death)	N/D
			M, 55	Pneumonia	CR	375 mg/8 h (12)	F	N/D
			M, 39	Pneumonia	CR	1.5 g/8 h (7)	F (death)	C
			M, 66	Pneumonia	CR	375 mg/8 h (16)	S	N/D
			F, 84	Pneumonia	CR	375 mg/8 h (8)	S	N/D
			M, 67	Pneumonia	CR	375 mg/8 h (14)	S	N/D
			M, 63	Pneumonia	CR	375 mg/12 h (16)	S	N/D
			M, 61	Pneumonia	CR	375 mg/8 h (5)	F (death)	N/D
			M, 71	Pneumonia	CR	3 g/8 h (5)	F	N/D
			F, 61	Pneumonia	CR	3 g/8 h (22)	S	N/D
			M, 64	Pneumonia	CR	3 g/8 h (14)	S	N/D
**Plant et al., 2018** [36]	Case report	1	M, ?	Pneumonia	MDR	1.5 g/8 h (?)	F (death)	C
**Romano et al., 2020** [37]	Case report	1	F, 63	Pneumonia	MDR	3 g/8 h (14)	S	N/D
**Soliman et al., 2015** [38]	Case report	1	M, 59	Pneumonia	PDR	3 g/8 h (14)	S	N/D
**Stokem et al., 2018** [39]	Case report	1	F, 35	Pneumonia	MDR	3 g/12 h (14)	S	C
**Vickery et al., 2016** [40]	Case report	1	M, 25	Pneumonia	MDR	3 g/8 h (12)	S	C
**Xipell et al., 2018** [41]	Observational study	8	M, 78	Pneumonia	MDR	?	S	E
			M, 69	Pneumonia	XDR	1.5 g/8 h (6)	F (death)	C
			M, 52	Pneumonia	MDR	1.5 g/8 h (8)	S	N/D
			M, 77	Pneumonia	XDR	3 g/8 h (3) → 2g/8h (3)	S	N/D
			F, 49	Tracheobronchitis	XDR	1.5 g/8 h (8)	S	E
			M, 49	Tracheobronchitis	MDR	?	S	E
			M, 64	Tracheobronchitis	XDR	1.5 g/8 h (7)	F	E
			M, 61	Tracheobronchitis	MDR	1.5 g/8 h (13)	S	E
**Zikri et al., 2019** [42]	Case report	1	F, 14	Pneumonia	MDR	1.5 g/8 h (?)	S	N/D

M, male; F, female; MDR, multidrug-resistant; XDR, extensively drug-resistant, PDR, pandrug-resistant; CR, carbapenem-resistant; S, success; F, failure; E, eradication; C, colonization; N/D, no data.

**Table 2 life-11-00474-t002:** Comorbidities.

Comorbidities	Cases (N=)
**Cystic fibrosis**	13
**COPD**	7
**Other respiratory diseases**	6
**Respiratory failure** − Requiring tracheotomy	15 9
**Hypertension**	8
**Other cardiac diseases**	12
**Diabetes**	7
**Acute kidney disease**	3
**Chronic kidney disease**	7
**Immunosupression conditions** − AIDS/HIV − CID − Neutropenia − N/D	7 2 2 1 2
**Solid cancer** − Lung − Rectum − Esophagus − Vulva − Breast − Bladder − Medulloblastoma	9 2 2 1 1 1 1 1
**Blood cancer** − Lymphoma − Leukemia − Myelodysplasia	9 3 4 2
**Transplantation** − Lung − Kidney − Heart − Liver	15 10 3 1 1
**Dementia**	4
**Quadriplegia/tetraplegia**	5
**Other neurological diseases**	8
**Alcoholism/cirrhosis**	7
**Other gastrointestinal diseases**	4

**Table 3 life-11-00474-t003:** Infection site and microorganisms involved.

Author, Year	Infection Site (Biological Sample)	Microorganisms
**Alessa et al., 2018**	Lungs (N/D)	*Pseudomonas Aeruginosa*
**Alqaid et al., 2015**	Urinary tract (culture)	Proteus mirabilis, Providencia stuartii
**Álvarez Lerma et al., 2017**	Lungs (BAS)	*Pseudomonas Aeruginosa*
	Lungs (N/D)	*Herpes simplex*
	Colon (N/D)	*Clostridium difficile*
**Ang et al., 2016**	Blood (catheter culture)	*Staphylococcus aureus*
**Dinh et al., 2017**	N/D	*Stenotrophomonas maltophilia*
	N/D	*Enterococcus faecalis*
	N/D	*K. pneumoniae*
	N/D	*Citrobacter koseri*
	N/D	*P. aeruginosa*
**Gelfand et al., 2015**	N/D	*Clostridium difficile*
**Haidar et al., 2018**	Lungs (N/D)	*MRSA*
	Lungs (N/D)	*MRSA*
	Lungs (N/D)	*MRSA*
	Lungs (N/D)	*Serratia marcescens*
	Blood (N/D)	*Vancomycin-resistant Enterococcus faecium*
	Blood (N/D)	*Vancomycin-resistant Enterococcus faecium*
	Abdomen (wound culture)	*Citrobacter fruendii*
	Blood (N/D)	*Candida tropicalis*
**Hernandez-Tejedor et al., 2016**	Blood (N/D)	*Pseudomonas Aeruginosa*
**Lewis et al., 2018**	Colon (N/D)	*Clostridium difficile*
	Lungs (N/D)	*Acinetobacter baumaunii*
**Maniara et al., 2021**	Blood (culture)	*Staphylococcus epidermidis*
	Lungs (N/D)	*Acinetobacter baumannii-calcoaceticus*
	Blood (culture)	*Candida auris*
**Munita et al., 2017**	Lungs (N/D)	*K. pneumoniae*
	Lungs (N/D)	*K. pneumoniae*
**Romano et al., 2020**	N/D	*MSSA, Pseudomonas aeruginosa*
**Soliman et al., 2015**	Lungs (N/D)	*MRSA, Pseudomonas aeruginosa*
**Xipell et al., 2018**	Lungs (BAS)	*MRSA*
	Lungs (BAS)	*Enterobacter cloacae*

MRSA, methicillin-resistant Staphylococcus aureus; MSSA, methicillin-sensitive Staphylococcus aureus; BAS, bronchial aspirate sample; N/D, no data.

## Data Availability

Data were collected in the Department of Women, Child, General and Specialistic Surgery, University of Campania “L. Vanvitelli”, Naples, Italy.

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
