# Peer review of "Ceftolozane/Tazobactam for Resistant Drugs Pseudomonas aeruginosa Respiratory Infections: A Systematic Literature Review of the Real-World Evidence"

_life, 2021, doi:10.3390/life11060474_

Round 1
Reviewer 1 Report
I have annotated some changes in the pdf itself

Author Response
Reviewer 1
- further data are necessary → further data is necessary
- Objectives → reorganized
- “A systematic review was performed based on the PRISMA (Preferred Reporting Items for Systematic Reviews and Meta-Analyses) guidelines [19]. The protocol was not published, but available on request. The review was not registered with the International prospective register of systematic reviews (PROSPERO), due to the long waiting times for the current situation.”
- The most common comorbidities were respiratories → The most common comorbidities were respiratory...

Reviewer 2 Report
In Table 1 some microorganisms are defined as MDR, XDR and PDR. The definitions are missing and should be included in “Materials and Methods” also with bibliographic reference. On the basis of these classifications, some considerations have been made on the therapeutic failure of C / T in the “Results” (page 10; lines 217-219) which require a correct indication of the definitions of the different resistance scenarios.
In the same table 1 some microorganisms are defined as "carbapenem-resistant": in the papers examined in the review were the "carbapenem-resistant" microorganisms distinguished from the "carbapenemase-producing" ones? Were there cases of KPC or other types of Gram-negative resistance (VIM, NDM)? This is important to better distinguish the different therapeutic response to C / T therapy.
The paper refers to the condition of "difficult-to-treat scenario" and "difficult-to-treat pulmonary PA infections": what are the conditions that define this scenario? What are the bibliographic references that define it? There is no definition or bibliographic reference in the Materials and Methods.
The definition of colonization made in table 1 on what parameters is defined? Colonization is not a clinical condition requiring antibiotic therapy, only a condition requiring surveillance in an antimicrobial stewardship program. Above all, it cannot be considered an outcome of therapy.
Among the comorbidities, co-infections were considered without a particular study on the etiology according to the affected apparatus. Describing that 17 organisms have been isolated in the lung is of no value. What are the isolated microorganisms in the lung? Which ones in the urinary tract? Which ones in the blood? And of these on which biological samples were they isolated? Sputum? BAL? tracheoaspirate? Peripheral vein blood? From CVC? Urinary catheter?
Lines 197-198: was the empirical therapy performed in 40 cases based on the absence of microbiological isolation? On what parameters? Radiological? Clinical?
What antibiotics were associated with C / T? for which proven or suspected infections? For which clinical conditions? All these data should then be analyzed in "Discussion".
Some comorbidities have a different weight and deserve a deepening in the "Discussion". Pseudomonas aeruginosa is one of the main and most frequent Gram-negatives responsible for serious infections in patients with cystic fibrosis, unlike in patients with organ transplantation.
In "Discussion" some more in-depth considerations should be made on the different conditions that determined the different dosage of C / T, the different outcome, the presence of comorbidities. It would be useful to supplement Table 1 with Table 2.
In general:
- correct the names of the microbiological species all in italics.
- Correct "organismis" with "microorganisms"
- it is more correct to write "Pseudomonas aeruginosa" and not "Pseudomonas Aeruginosa"
Author Response
- MDR, XDR and PDR definitions missing → added: “According to an international expert proposal, MDR is defined as “non-susceptibility to at least one agent in three or more antimicrobial categories”; XDR is defined as “non-susceptibility to at least one agent in all but two or fewer antimicrobial categories (i.e. bacterial isolates remain susceptible to only one or two categories)”; PDR is defined as “non-susceptibility to all agents in all antimicrobial categories (i.e. no agents tested as susceptible for that organism)”
- "carbapenem-resistant" → added: “Among CR-PA, no cases of Carbapenemase Producing Enterobacteriaceae (CPE) were reported. ”
- "difficult-to-treat scenario" and "difficult-to-treat pulmonary PA infections" → deleted.
- colonization in table 1 → modified: “Microbiological status”
- comorbidities → table 3 added
|
Author, year |
Infection site (biological sample) |
Microorganisms |
|
Alessa AM et al, 2018 |
Lungs (N/D) |
Pseudomonas Aeruginosa |
|
Alqaid MM et al, 2015 |
Urinary tract (culture) |
Proteus mirabilis, Providencia stuartii |
|
Álvarez Lerma F et al, 2017 |
Lungs (BAS) |
Pseudomonas Aeruginosa |
|
Lungs (N/D) Colon (N/D) |
Herpes simplex Clostridium difficile |
|
|
Ang JY et al, 2016 |
Blood (catheter culture) |
Staphylococcus aureus |
|
Dinh A et al, 2017 |
N/D |
Stenotrophomonas maltophilia |
|
N/D |
Enterococcus faecalis |
|
|
N/D |
K. pneumoniae |
|
|
N/D |
Citrobacter koseri |
|
|
N/D |
P. Aeruginosa |
|
|
Gelfand MS et al, 2015 |
N/D |
Clostridium difficile |
|
Haidar G et al, 2018 |
Lungs (N/D) |
MRSA |
|
Lungs (N/D) |
MRSA |
|
|
Lungs (N/D) |
MRSA |
|
|
Lungs (N/D) |
Serratia marcescens |
|
|
Blood (N/D) |
Vancomycin-resistant Enterococcus faecium |
|
|
Blood (N/D) Abdomen (wound culture) Blood (N/D) |
Vancomycin-resistant Enterococcus faecium Citrobacter fruendii Candida tropicalis |
|
|
Hernandez-Tejedor A et al, 2016 |
Blood (N/D) |
Pseudomonas Aeruginosa |
|
Lewis PO et al, 2018 |
Colon (N/D) Lungs (N/D) |
Clostridium difficile Acinetobacter baumaunii |
|
Maniara BP et al, 2021 |
Blood (culture) Lungs (N/D) Blood (culture) |
Staphylococcus epidermidis Acinetobacter baumannii-calcoaceticus Candida auris |
|
Munita JM et al, 2017 |
Lungs (N/D) |
K. pneumoniae |
|
Lungs (N/D) |
K. pneumoniae |
|
|
Romano MT et al, 2020 |
N/D |
MSSA, Pseudomonas Aeruginosa |
|
Soliman R et al, 2015 |
Lungs (N/D) |
MRSA, Pseudomonas Aeruginosa |
|
Xipell M et al, 2018 |
Lungs (BAS) |
MRSA |
|
Lungs (BAS) |
Enterobacter cloacae |
|
|
MRSA, Methicillin-Resistant Staphylococcus Aureus; MSSA, Meticillin Sensitive Staphylococcus Aureus; BAS, bronchial aspirate sample; N/D, no data. |
||
- empirical therapy → added: “empirical treatment was performed in 40 cases in the absence of microbiological isolation based on clinical or radiological parameters.”
- antibiotics associated with C / T → added “The most common association was with colistin intravenously or inhaled (27 patients in Alqaid MM et al, Bosaeed M et al, Castón JJ et al, Dinh A et al, Haidar G et al, Lewis PO et al, Maniara BP et al, Munita JM et al), tobramycin intravenously or inhaled (15 patients in Alessa AM et al, Haidar G et al, Lewis PO et al, Munita JM et al, Vickery S et al, Xipell M et al), ciprofloxacin intravenously (7 patients in Ang JY et al, Haidar G et al, Romano MT et al, Vickery S et al), amikacin intravenously (5 patients in Dinh A et al, Xipell M et al, Zikri A et al), and linezolid intravenously (4 patients in Haidar G et al, Soliman R et al).”
- comorbidities / serious infections in patients with cystic fibrosis, unlike in patients with organ transplantation → added “According to literature, PA cause infections in patients with comorbidities, such as: cystic fibrosis, chronic structural lung disease (e.g., bronchiectasis, COPD), impaired immune defenses (e.g., HIV infection, malignancy with neutropenia or recent chemotherapy, organ transplant recipients), diabetes mellitus, reinal failure and hemodialysis, chronic cardiovascular or neurological disease.”
- considerations dosage of C / T, the different outcome, the presence of comorbidities → added “Ceftolozane has shown adequate penetration in lung tissue, close to 50%, with 2 g doses in an extended infusion of 3 hours”, “The increased dose of C/T could be associated with an increase in the occurrence of adverse effects, such as elevation of liver enzymes, hypotension, or cutaneous reactions”
- correct the names of the microbiological species all in italics → done.
- correct "organismis" with "microorganisms" → done.
- write "Pseudomonas aeruginosa" and not "Pseudomonas Aeruginosa" → done.

Reviewer 3 Report
I think the review is fine as the authors clearly comment on the limitation of putting together a lot of case series. However, this is how medicine works - if you are standing with the back against the wall you take every hint - and this is the C/T combination in the described situation although it was approved for a different situation. In my opinion they should add a short comment on off-label use of C/T in the described circumstances - is that true worldwide or how is the exact status in the analyzed countries ?
The conclusion therefore is correct: It may have an advantage in the described situation which has to be proven.
Well done, congrats.
Author Response
Off-label use of C/T is that true worldwide.

Round 2
Reviewer 2 Report
Authors responded to all previous review requests.
It is accepted without any further revision.